# Towards Convolutional Neural Network Acceleration and Compression Based on *Simon*
*k*-Means

**DOI:** 10.3390/s22114298

**Published:** 2022-06-06

**Authors:** Mingjie Wei, Yunping Zhao, Xiaowen Chen, Chen Li, Jianzhuang Lu

**Affiliations:** The College of Computer Science, National University of Defence Technology, Changsha 410000, China; weimingjie20@nudt.edu.cn (M.W.); zhaoyunping@nudt.edu.cn (Y.Z.); xwchen@nudt.edu.cn (X.C.); lichen@nudt.edu.cn (C.L.)

**Keywords:** convolutional neural networks, deep learning, k-means, model compression, weight quantization

## Abstract

Convolutional Neural Networks (CNNs) are popular models that are widely used in image classification, target recognition, and other fields. Model compression is a common step in transplanting neural networks into embedded devices, and it is often used in the retraining stage. However, it requires a high expenditure of time by retraining weight data to atone for the loss of precision. Unlike in prior designs, we propose a novel model compression approach based on *Simon*
*k*-means, which is specifically designed to support a hardware acceleration scheme. First, we propose an extension algorithm named *Simon*
*k*-means based on simple *k*-means. We use *Simon*
*k*-means to cluster trained weights in convolutional layers and fully connected layers. Second, we reduce the consumption of hardware resources in data movement and storage by using a data storage and index approach. Finally, we provide the hardware implementation of the compressed CNN accelerator. Our evaluations on several classifications show that our design can achieve 5.27× compression and reduce 74.3% of the multiply–accumulate (MAC) operations in AlexNet on the FASHION-MNIST dataset.

## 1. Introduction

Convolutional neural networks have been some of the most successful machine learning techniques in the last decade [1]. They are widely used in many applications, such as autonomous driving [2], automatic speech recognition [2], and weather forecasting. The inference-time latency and energy efficiency of CNNs are the key assessment indicators. In order to solve the problem of the excessive computational resource overhead of a CNN, some designs have proposed solutions based on Graphics Processing Units (GPUs), Field-Programmable Gate Arrays (FPGAs) [3,4,5], Application-Specific Integrated Circuits (ASICs) [6], and other hardware to facilitate the acceleration of CNNs. However, GPUs are impacted by a high power consumption problem, while ASICs are affected by high production and development costs, which greatly limit their scope of application. This research [7] can be roughly divided into two directions. One of the directions is accelerating the multiplication process in convolution and fast and efficient parallel computing [8]. The other direction is model compression, which reduces the model size while maintaining the same accuracy or loss [9] to reduce multiply–accumulate operations and storage overhead. The current model compression is in the retraining stage, and fine-tuning is used to compensate for the loss of accuracy, but this will add additional retraining time, and pre-training is not applicable in some cases.

Inspired by this, this paper proposes a model compression scheme for the trained weight, which can significantly reduce the number of multiply–accumulate operations and minimize the scale of the model with a slight loss of accuracy. The main contributions of this paper are as follows:We propose a novel compression approach named *Simon*
*k*-means to cluster trained weights, which is based on simple *k*-means. We use *Simon*
*k*-means to cluster trained weights in convolutional layers and fully connected layers to minimize the neural network models. The compressed models can meet the requirements of applications in embedded or mobile devices. To the best of our knowledge, ours is the first work to compress the data after training weights.We provided a hardware implementation of the compressed CNN accelerator. We optimized the data flow to make the most of the on-chip data reuse, which can reduce the access to the off-chip storage. In addition, we used the data storage and index approach to take full advantage of the compressed model’s data characteristics. We showed that our approach can reduce the consumption of hardware resources in data movement and storage.We mapped several CNN workloads to the proposed architecture. Our evaluations show that our design can achieve 5.27× compression and reduce the MAC operations in AlexNet to 74.3% on the FASHION-MNIST dataset.

The other parts of this paper are organized as follows: In Section 2, a brief introduction to CNNs and the *k*-means algorithm is given. Section 3 presents and discusses the related literature. Section 4 explains the *Simon*
*k*-means algorithm and the accelerator architecture design. Section 5 compares and analyzes the experimental results. Section 6 summarizes the entire work.

## 2. Background

### 2.1. CNN

CNNs generally consist of an input layer, convolutional layer, ReLU layer, pooling layer, and fully connected layer [1]. The key operation of a CNN is convolution; the input of the convolutional layer is an input feature map of $H_{In}$ × $W_{In}$ × $C_{In}$ and the $M_{Out}$ kernel of K × K × $C_{In}$; then, the stride S and padding method for each movement of the convolution kernel are set, and a convolution operation is performed to generate an output feature map of $H_{Out}$ × $W_{Out}$ × $M_{Out}$. Most CNNs have multiple convolutional layers. The convolutional layers tend to require most of the computing resources and time in the training and inference [9]. As shown in Figure 1, the convolutional layers occupy most of the computing resources and time, and most of the parameters are concentrated in the fully connected layer. Therefore, the acceleration of the convolutional layer is the key to accelerating the neural network, and the compression of the fully connected layer is the key to minimizing the model. Therefore, the paper focuses on the acceleration of the convolutional layer and the compression of the fully connected layer.

### 2.2. Model Compression

CNNs cost a lot of computing and storage resources, so it is difficult to apply them to some embedded systems with limited hardware resources [10]. Model compression is currently the most popular method for this problem. It is committed to accelerating inference speed and reducing storage size while maintaining the original characteristics and accuracy of the model. Current model compression strategies can be divided into four categories: pruning and weight sharing, quantization, knowledge distillation, and low-rank factorization. Pruning uses an effective evaluation method for pruning unimportant connections and filters to simplify the trained network model [11]. The quantization reduces the size of the network by using lower-weight bits [10]. Knowledge distillation puts forward a teacher–student network, which extracts useful information from the cumbersome network (teacher network) and migrates it to the distilled network (student network). The distilled network can have performance similar to that of the cumbersome network, and the number of calculations is also reduced [12]. The idea of low-rank factorization is to treat the convolution kernel as a tensor in four dimensions and remove its redundancy based on the tensor decomposition to improve acceleration [13].

### 2.3. K-means Algorithm

*K*-means is a classic clustering algorithm. From the given $x_{1}$, $x_{2},\cdots$$x_{n}$, we select *k* centroids $c_{1}$, $c_{2},\cdots$, $c_{k}$ to represent these data and minimize the sum of the distance between the centroid and each datum. Equation (Equation 1) describes the goal of the *k*-means algorithm, which is to minimize the Euclidean distance between the centroid and each datum [14].
(1)$$Dis=\sum_{j=1}^{k}\sum_{i=1}^{n}{\|x_{i}^{\left(j\right)}-c_{j}\|}^{2}.$$

## 3. Related Work

On the hardware acceleration front, researchers increase throughput through loop unrolling, reuse, parallel computing, the tiling factor, etc. [2,4,5]. Nihat proposes general reuse and a reuse-center CNN accelerator [2]. Yufei deeply analyzes the convolutional cycle acceleration strategy by characterizing the loop optimization techniques [5]. Chen Zhang proposes a roofline-model-based method to optimize the CNN’s computation and memory access [4].

Zhao proposes a dynamically reconfigurable accelerator architecture that implemented a Sparse–Winograd F(3×3,2×2)-based high-parallelism hardware architecture [15]. Based on the Sparse–Winograd algorithm, he proposed a method for decomposing convolution based on F(3×3,2×2), which eliminated the complex pre-operation of the Winograd algorithm, reduced the difficulty of the hardware implementation of the algorithm, and greatly expanded the hardware flexibility. The main consideration for the decomposition with F(3×3,2×2) as the basis is that the parameters in the transformation matrices A, B, and G are (0,±1,±1⁄2), which can be easily implemented in hardware by shifting or adding hardware, which is not only simple and easy to control, but also reduces hardware expenses and reduces the design power consumption and cost [16].

Model compression reduces a CNN’s calculation and storage overhead by compressing the convolution kernel. At present, most model compression is based on the Pruning, Trained Quantization, and Decoding proposed in [10], which compressed AlexNet by 35× and VGG-16 by 49×. The clustering method in [10] is based on *k*-means; the fine-tuning step was applied to the complete set of weights to compensate for the loss in the accuracy due to the clustering of the weights. In addition, due to the book-keeping, encoding, and compression scheme, deep compression is not favorable for hardware acceleration and aimed at the retraining stage, but the overall training time was basically the same as the training time without the pre-weight [10].

The authors discuss the effectiveness of pruning [11]. Yiwen Guo proposes dynamic model-pruning methods, including pruning and splicing, where pruning refers to cutting off unimportant weights, but the basis for determining the importance of weights is not intuitive [17]. Therefore, splicing was added, which could repair important but pruned weights. Li proposes a pruning method based on the magnitude and determined the pruning filter according to the mean value of the filter weight [18]. Hu defines an APoZ (Average Percentage of Zeros), and the proportion of the value of activation of zeros in each filter was used as the benchmark for pruning [19]. Yang proposes a pruning method based on energy consumption, which evaluates the energy consumption of each layer of the model and prioritized the pruning of the layers with higher energy consumption [20]. An interesting pruning method was given in [21], where the author uses random pruning and then calculated the performance of the model to determine a locally optimal pruning scheme.

The author of [22,23] replaces the traditional single-precision floating-point data with quantized weight data as a fixed point. Using low-precision fixed-point numbers instead of high-precision floating-point numbers to perform calculations can significantly improve energy consumption and throughput, but it will inevitably cause a loss of accuracy. Early network models, such as LeNet-5, have fewer convolutional layers, and the loss of accuracy is acceptable. However, with the emergence of cumbersome neural networks, such as mobilenet and resnet, the stochastic rounding scheme in [22] does not guarantee that the loss of accuracy is still within an acceptable range.

Hinton defines knowledge distillation, which extracts useful information from a cumbersome network and migrates it to a smaller network. The small learned network can have a performance close to that of the cumbersome network and can greatly save computing resources [12]. Zagoruyko draws on the idea of distilling by using an attention map that could provide visually relevant location information in the complex network to supervise the learning of the small network and combining the three-level features of low, medium, and high [24].

    Max proposes a linear-combination-convolution-kernel-based method using n×1+1×n convolution kernels instead of n×n convolution kernels to perform low-rank approximation, achieving a 4.5× speedup with less than 1% drop in accuracy [13]. Kim presents a Tucker decomposition called one-shot whole-network compression, consisting of three steps: rank selection with variational Bayesian matrix factorization, Tucker decomposition on a kernel tensor, and fine-tuning to recover accumulated loss of accuracy [25]. Wenqi shows that tensor ring networks compress the convolutional and fully connected layers of deep neural networks [26]. Jinmian uses block–term tensor decomposition to compress recurrent neural networks [27].

    A recent development is INQ (incremental network quantization), which was proposed in [28], consisting of three steps: weight partition, weight quantization, and retraining. First, a measurement method is used to divide the weights in each layer of the pre-trained CNN model into two disjoint groups, the weight of the first group is quantified, and, finally, the weight of the other group is fine-tuned to compensate for the loss of accuracy. This process is repeated until all weights are quantified. However, the quantitative weight value was limited to $2^{n}$, resulting in a large deviation between the quantized weight value and the original weight value, which would increase the retraining time. Akshay further improved on the basis of INQ [9], and the weight value was no longer limited to $2^{n}$. Instead, *k*-means was used for clustering to quantify weights, and *Symmetric*
*k*
-means were proposed; then, it was only necessary to find half of the centroid, but Akshay did not apply this to the compression of the fully connected layers.

The current quantization-based compression methods basically rely on retraining and fine-tuning for accuracy compensation. As far as we know, we should be the first quantization compression scheme that improves the clustering algorithm for accuracy compensation.

## 4. Algorithm and Hardware Design

The top-level design is shown in Figure 2. The algorithm design can be regarded as a pre-processing operation for the trained weight. For the convolutional layers and the fully connected layers, the trained weights are clustered through *Simon*
*k*-means for convolutional layers and *Simon*
*k*-means for fully connected layers, respectively. Then, the centroids are encoded, and the weight matrix stores the index of each weight. For the hardware design, we provide the hardware implementation of the compressed CNN accelerator.

### 4.1. Quantization for Convolutional Layers Using Simon k-Means

The authors of [9] proposed a method of clustering to quantify weights in the retraining stage; in [9], the *k*-means were used three times, and fine-tuning was used two times to complete the training of the network. For a complete CNN with multiple convolutional kernels, this will consume a lot of computing resources. This paper inherits the idea of clustering to quantify weights from [9], but it is no longer used for weights in the retraining stage, but rather for trained weights. In other words, we propose a pre-processing algorithm for the trained weights before the inference stage. Figure 2 performs clustering to quantify the trained convolution kernel. Only one clustering is required for the weight of each layer, and the compensation for the loss of accuracy does not rely on the fine-tuning of retraining, but on the clustering algorithm. Simple *k*-means and variants cannot satisfy our requirement of using one clustering operation to compress the model. Based on *k*-means, we have facilitated the selection of the initial centroid and the update of the centroid. The specific algorithm is shown in Algorithm 1.
**Algorithm 1:***Simon**k*-means For Convolutional Layers.**Require:** Input Matrix x with dimension K × K**Ensure:** Output Matrix x with dimension K × K1:$Initialize\;with\;cluster=\left[\left[\right]\right],means=\left[\right],cluster_{mean}=\left[\right],weight=flatten\left(x\right)$2:weight = sort (weight)3:**for** k of *K* **do**4:      **for** i of k **do**5:            sum = sum + weight [*k* × K + i]6:      **end for**7:      means.append (sum ÷ *k*)8:      sum = 09:**end for**10:cluster = find_cluster (means, weight, k)11:**for**num of weight **do**12:      **for** clu of cluster **do**13:            **if** num in clu **then**14:                 num = sum (clu) ÷ len (clu)15:            **end if**16:      **end for**17:**end for**

The solution to the initial centroid selection is reflected in lines 2–8. First, the weights are sorted. After dividing the weights into *k* groups, there are *k* weights in each group; we find the average number to obtain the initial centroid. The traditional *k*-means algorithm updates the centroid after dividing a piece of data for updating centroids. For a K×K convolution kernel, obtaining the final centroid requires K×K divisions. For the updated centroid of *Simon*
*k*-means for convolutional layers, after each division, we take the average of each cluster as the final centroid so that we only use *K* divisions. The method of dividing clusters is the same as *k*-means, which calculates the distance between the data and each centroid, then finds the minimum distance and classifies it into the corresponding cluster.

As shown in Figure 3, the weight matrix stores 32-bit floating-point numbers. The weight matrix will encode weights after clustering and stores the index of each weight. The number of bits of the weight index is much less than 32 bits, so the purpose of compression can be achieved by storing the weight index. The formula for the compression ratio of the convolutional layer is as follows:(2)$$r=\frac{N_{c}\times \log_{2}k+k\times b}{N_{c}\times b},$$
where *r* represents the compression ratio, $N_{c}$ represents the number of weights in the convolutional layer, k represents the number of clusters, and b represents the bits of the weight (float32).

### 4.2. Hardware Acceleration Strategy

In Figure 4, we give an example to explain the hardware acceleration principle of our acceleration architecture to reduce multiply–accumulate operations in order to accelerate convolution. First of all, we compress the convolution kernel through clustering to quantify and encode for compression. According to the traditional operation, we perform convolution operations on the input feature map and the corresponding weights, which require 9 multiplications and 8 additions as shown in Figure 4a. We accumulate the data corresponding to the same cluster centroid position in the input feature map through the accumulator and, finally, perform multiply–accumulate operations with the cluster centroid, which only needs 3 multiplications and 8 additions as shown in Figure 4b. Assuming that each layer in the convolution kernel is 3×3, the number of multiply–accumulate operations is reduced by 2/3; in fact, a convolutional layer with a convolution kernel size of 7×7×3×64 appears in ResNet50, which further reduces the number of multiply–accumulate operations. The traditional algorithm of the CNN accelerator is shown in Algorithm 2, and the CNN accelerator using the acceleration strategy needs to replace part of the multiply–accumulate operations with the accumulator. The specific algorithm is shown in Algorithm 2. 
**Algorithm 2:** Convolution Computation with Quantized Weight.**Require:** Input Feature Map and quantized weight filters**Ensure:** Output Feature Map1:**for**row of Input Feature Map **do**2:      **for** column of Input Feature Map **do**3:            **for** num of weight filters **do**4:                 Acc[] = 05:                 **for** channel of Input Feature Map **do**6:                       **for** number of quantized weight **do**7:                             **for**
*k* of clusters **do**8:                                   Acc[k] += input9:                                   Output += Acc[k] × Centroid[k]10:                           **end for**11:                     **end for**12:               **end for**13:          **end for**14:    **end for**15:**end for**

### 4.3. Quantization for Fully Connected Layers Using Simon k-Means

Akshay explained that the weights of the convolutional layers are almost mirrored distributions about zero [9]. An interesting finding is that the weights of each fully connected layer are also almost mirrored distributions with respect to zero. Figure 5 shows the histograms of the weight distributions of AlexNet, ResNet50, and LeNet-5 on FASHION-MNIST. There are 11 bins for each layer, and it can be observed that each fully connected layer basically satisfies the mirror distribution with respect to zero. Inspired by this, we present *Simon*
*k*-means for fully connected layers to compress the fully connected layers.

In *Simon*
*k*-means for fully connected layers, our goal is still to find *k* centroids. According to the symmetry of the data, we only need to calculate k/2 centroids through the objective formula and then multiply it by −1 to get all of the centroids.
(3)$$Dis=\sum_{j=1}^{k/2}\sum_{i=1}^{n}{\|x_{i}^{\left(j\right)}-c_{j}\|}^{2},$$

For *k*-means, each fully connected layer needs to store *k* 32-bit floating-point centroids. For *Simon*
*k*-means for fully connected layers and k/2 effective centroids, only k/2 32-bit floating-point centroids need to be stored, and the model is further compressed. Compared to *k*-means, *Simon*
*k*-means for fully connected layers has a faster convergence speed, and the compression efficiency is further improved. The formula for the compression ratio of the fully connected layer is as follows:(4)$$r=\frac{N_{f}\times \log_{2}k+k/2\times b}{N_{f}\times b},$$
where r represents the compression ratio, $N_{f}$ represents the number of weights in the fully connected layer, *k* represents the number of clusters, and b represents the bits of the weight (float32). There is no loss of precision at 5-bit quantization in *Simon*
*k*-means for fully connected layers compared to 32-bit floating-point baseline models.

### 4.4. The Overview of the Accelerator Architecture

In order to maximize the performance of the hole design, we propose a hardware accelerator architecture design according to the characteristics of the data processing. Figure 6 shows the top-level diagram of the accelerator design. We use a three-level cache design to complete the data transmission. Due to the constraint of storage space, the input images and weights are initially stored in DRAM. We chunk the data and move them from DRAM to SRAM, which helps us reduce the hardware storage consumption. After transferring the data from SRAM to the input buffer, the tiles of the input feature map and corresponding weight index matrix will be transmitted to the accumulator. Unlike in the input feature map, the weights will be directly transmitted to the PE array. Then, the PE array will perform the computation of the dot product between weights and the result of the accumulation. Finally, the final data, which are the results of the summation between different channels, will be returned to the external memory with the help of the output buffer.

All layers of the neural network are modeled as different states of the finite-state machine and sent to the accelerator in turn. Some blocks of the input feature map and their corresponding weights in this convolutional layer are fetched from the RAM and are stored in the on-chip buffer. The computational unit performs the convolution operation, bias function, and activation function. One layer of the convolutional layer operation is performed each time, and then the corresponding intermediate output results are sent back to the RAM. This data flow is followed until all convolutional layers and pooling layers are computed. Finally, the output feature map is loaded into the CPU for the calculation of the fully connected layer to obtain the final prediction result.

## 5. Results and Discussion

To evaluate the compression effect, we conducted a large number of network classification tasks on Dogs-Vs-Cats, MNIST, FASHION-MNIST, and CIFAR10. Dogs-Vs-Cats is a simple dataset with two categories that distinguish dogs and cats. The MNIST is a relatively simple handwritten digit subclass dataset. The FASHION-MNIST is an improvement of MNIST, which uses ten kinds of clothes instead of handwritten numbers, representing difficult subclass situations. CIFAR10 has about 50,000 training images and 10,000 validation images. Each image is annotated as one of 10 object classes. We applied our compression scheme to AlexNet, LeNet-5, ResNet-50, and ResNet-101, covering almost all known deep CNN architectures.

### 5.1. The Compression of Convolutional Layers

In order to evaluate the effect of *Simon*
*k*-means, we evaluated three popular convolutional neural networks in the “Dogs-Vs-Cats, MNIST, FASHION-MNIST” dataset. The uncompressed network model on the dataset was used as the baseline. The reductions of multiply–accumulate operations were compared by showing the compression before and after each convolution layer.

#### 5.1.1. Accuracy

We used LeNet-5, ResNet-101, and ResNet50 to evaluate the compression effects of convolutional layers. LeNet-5 has only two convolutional layers, and thus represents a simple convolutional neural network. The ResNet network parameters are primarily concentrated in the convolutional layer, making it easier to observe the compression effect. ResNet50 uses residual convolution and has more convolutional layers, representing a regular CNN. Resnet101 adds many blocks to conv4 of ResNet50, therefore representing a complex CNN. The accuracy indicators in Table 1 refer to Top-1 Accuracy. The results in Table 1 show the loss of accuracy caused by *Simon*
*k*-means compression with the three datasets of the three neural networks. It can be seen from the table that the loss of accuracy caused by *Simon*
*k*-means is very low, basically fluctuating at 1%, and the fluctuation obviously does not exceed the confidence interval.

#### 5.1.2. The Reduction of Multiply–Accumulate Operations

With the input image of 28×28×1 as the input of LeNet-5, it can be seen that the number of multiply–accumulate operations was significantly reduced. The results in Table 2 show the number of multiply–accumulate calculations for each convolutional layer of LeNet-5 before and after compression. After compression, the number of multiply–accumulate operations in each convolutional layer was reduced by 66.67%. With the input image of 224 ×224×3 as the input of ResNet50, the results in Table 2 show the number of multiply–accumulate calculations for the convolutional layers of ResNet50 before and after compression. After compression, the number of multiply–accumulate calculations in the first convolutional layer was reduced by 85.7%, and those in the other convolutional layers were all reduced by 66.67%. According to the data of AlexNet in the table, the reduction and compression ratios of the convolutional layers increased with the increment in convolutional kernels.

### 5.2. The Compression of Fully Connected Layers

In order to evaluate the compression effect of the fully connected layers, we compressed each fully connected layer of the weight of AlexNet on the FASHION-MNIST dataset and calculated the compression ratio and accuracy loss with different values of *k*. The uncompressed network model on the dataset was used as the baseline. The accuracy indicators in Figure 7 and Figure 8 refer to Top-1 accuracy.

#### 5.2.1. Accuracy

Figure 7 and Figure 8 show the changes in the accuracy of each fully connected layer in LeNet-5 and ResNet50 after using *k*-means and *Simon*
*k*-means for fully connected layers with different values of *k*. According to Figure 8, we can reach three conclusions. First, the loss of accuracy decreased gradually and tended to converge with the increment in *k*. Second, when *k* = 32, the loss of accuracy was close to zero. Third, there was no difference in the accuracy of *k*-means and *Simon*
*k*-means when *k* ≥ 16.

#### 5.2.2. Compression Ratio

Figure 9 shows the compression ratio for fully connected layers with different values of *k*. We can draw three conclusions: First, with the increment in *k*, the compression ratio gradually decreased. Secondly, *Simon*
*k*-means for fully connected layers showed a better compression effect than that of *k*-means. Third, *Simon*
*k*-means was more effective for small-scale fully connected layers on compression.

### 5.3. Discussion

Table 3 describes the loss of precision and the compression ratios of different compression methods. From Table 3, we can see that the compression ratio of the compression scheme in this paper is second only to those of XNOR-Net and TWN. However, XNOR-Net and TWN cause a greater loss of precision. Compared to other compression schemes, the method proposed in this paper has no absolute advantage in terms of compression ratio and loss of accuracy. The compression scheme proposed in this paper is suitable for situations in which a slight loss of precision can be accepted and those that require a higher compression ratio.

## 6. Conclusions and Future Work

This paper proposes a novel model compression algorithm based on *Simon*
*k*-means that is specifically designed to support hardware acceleration schemes. First, we propose an extended algorithm named *Simon*
*k*-means that is based on simple *k*-means. We use *Simon*
*k*-means to cluster trained weights in convolutional layers and fully connected layers. Then, we reduce the consumption of hardware resources in data movement and storage by using a data storage and index approach. Finally, we provide the hardware implementation of the compressed CNN accelerator. For fully connected compression, we achieve a compression ratio of 10.66× without loss of precision. We focus only on image classification in this paper; in the future, we will try our compression scheme on other CNN applications, such as object detection and depth estimation.

## Figures and Tables

**Figure 1 sensors-22-04298-f001:**
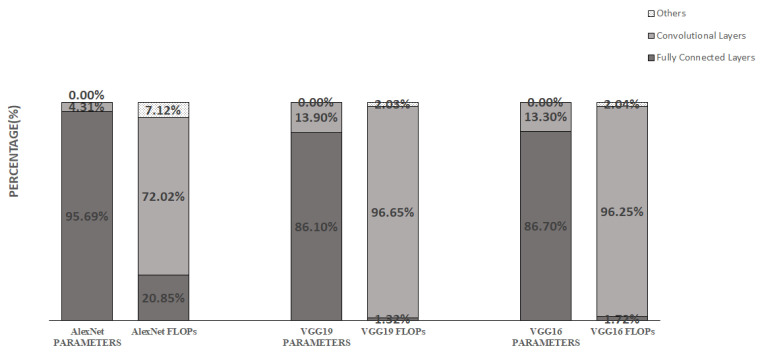
Profiling for convolutional neural networks.

**Figure 2 sensors-22-04298-f002:**
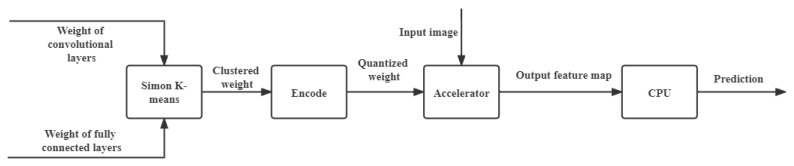
Overview of the design.

**Figure 3 sensors-22-04298-f003:**
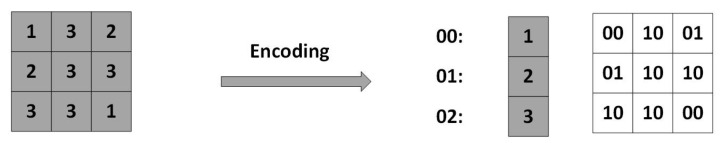
Example of encoding.

**Figure 4 sensors-22-04298-f004:**
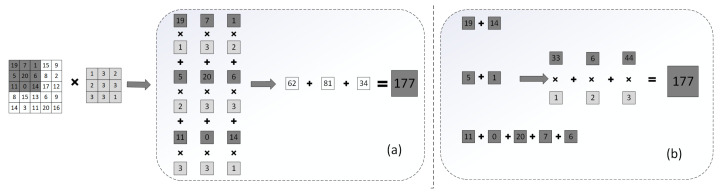
Example of the hardware acceleration strategy.

**Figure 5 sensors-22-04298-f005:**
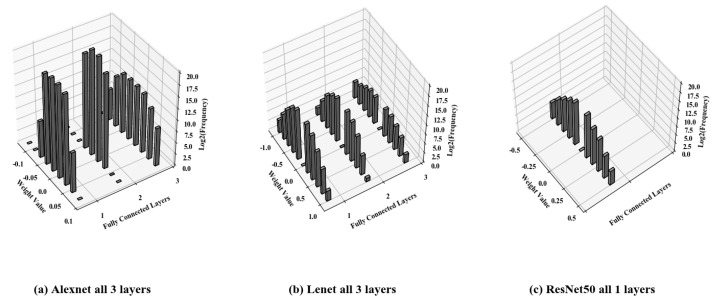
Histograms to demonstrate the symmetric nature of the weight distribution in fully connected layers on FASHION-MNIST.

**Figure 6 sensors-22-04298-f006:**
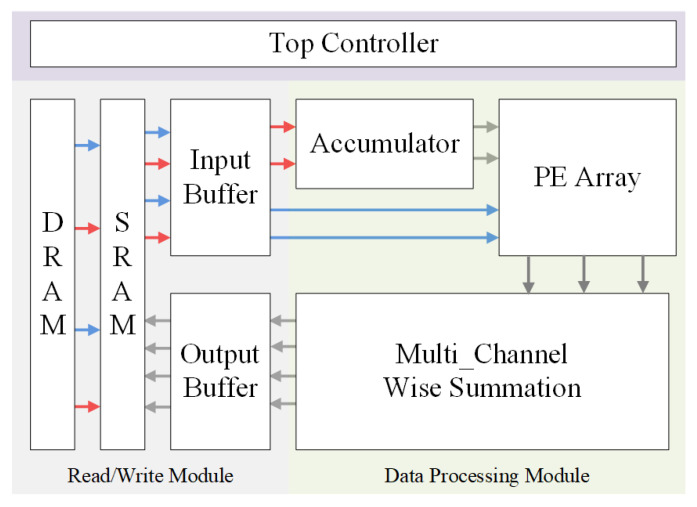
Hardware architecture: The red and blue arrows correspond to the data flow of the input images and weights.

**Figure 7 sensors-22-04298-f007:**
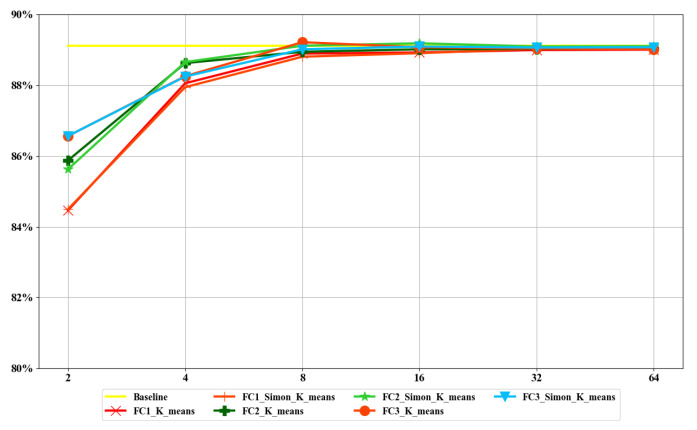
Accuracy of LeNet-5 for each fully connected layer with different values of *k*.

**Figure 8 sensors-22-04298-f008:**
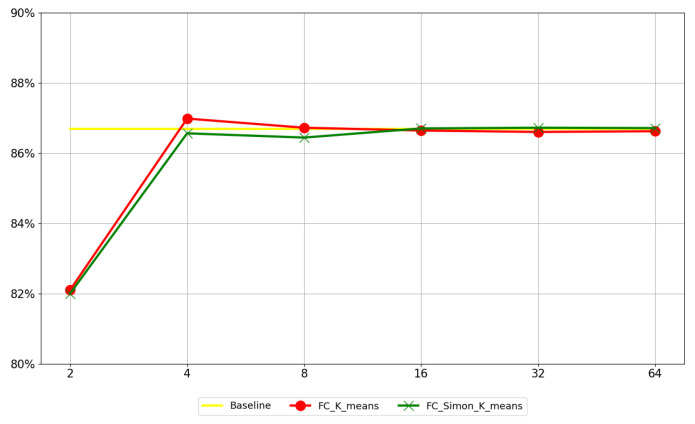
Accuracy of ResNet50 for each fully connected layer with different values of *k*.

**Figure 9 sensors-22-04298-f009:**
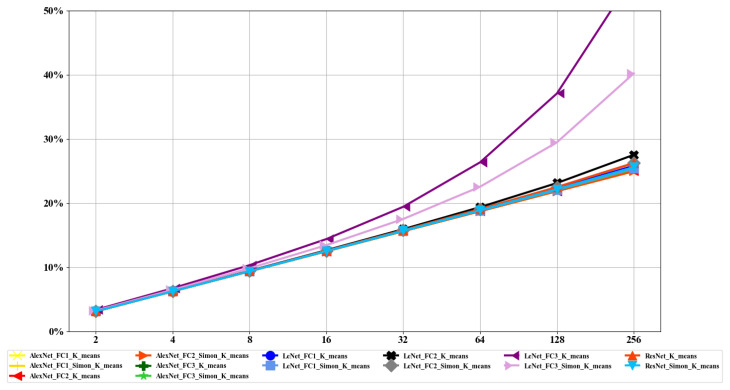
Compression ratio of fully connected layers of the neural network with different values of k.

**Table 1 sensors-22-04298-t001:** The accuracy of three CNNs with different datasets and with clusters for quantification.

Networks	Baseline	Compression	Dataset
LeNet-5	88.12%	87.20%	Cat vs. Dog
ResNet-50	98.78%	98.14%	Cat vs. Dog
ResNet-101	94.16%	93.64%	Cat vs. Dog
LeNet-5	99.02%	98.89%	MNIST
ResNet-50	98.46%	98.37%	MNIST
ResNet-101	97.93%	97.28%	MNIST
LeNet-5	89.12%	87.84%	FASHION-MNIST
ResNet-50	87.27%	85.80%	FASHION-MNIST
ResNet-101	87.43%	87.20%	FASHION-MNIST
AlexNet	89.82%	89.29%	FASHION-MNIST

**Table 2 sensors-22-04298-t002:** The number of multiply–accumulate calculations per inference for each convolutional layer of LeNet-5.

Networks	Convolutional Layer	Before Compression	After Compression	Reduction (%)	Compression Ratio of Convolutional Layer (%)
LeNet-5	***T*${}_{n}=$ 1**	117,600	23,520	80	29.37
LeNet-5	***T*${}_{n}=$ 2**	470,400	94,080	80	29.37
AlexNet	***T*${}_{n}=$ 1**	1,795,682,592	163,243,872	90.9	12.53
AlexNet	***T*${}_{n}=$ 2**	7,984,742,400	1,596,948,480	75	29.37
AlexNet	***T*${}_{n}=$ 3**	2,874,507,264	958,169,088	66.67	39.58
ResNet50	***T*${}_{n}=$ 1**	118,013,952	16,859,136	85.71	23.67
ResNet50	***T*${}_{n}=$ 2**	115,605,504	38,535,168	66.67	39.58

**Table 3 sensors-22-04298-t003:** Loss of precision and compression ratios of different compression methods in AlexNet.

Method	Accuracy Fluctuations (Top_1)	Compression Ratio	Dataset
SVD [29]	−2.02%	5×	IMAGENET
Symmetric *k*-means [9]	+0.04%	1.04×	IMAGENET
INQ [28]	−0.25%	1.04×	IMAGENET
XNOR-Net [30]	−12.32%	32×	IMAGENET
TWN [31]	−2.02%	16×	IMAGENET
Data-free pruning [32]	−1.40%	1.5×	IMAGENET
This paper	−0.83%	5.27×	FASHION-MNIST

## Data Availability

The data presented in this study are available on request from corresponding authors.

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
