# Peer review of "Towards Convolutional Neural Network Acceleration and Compression Based on Simonk-Means"

_sensors, 2022, doi:10.3390/s22114298_

Round 1

Reviewer 1 Report

The authors present a paper which probably contains a new and important result. However, the paper contains many errors which must be corrected before evaluation of the novelty and significance of the result.

English must be improved: even in the Abstract, there are "but" in the beginning of a sentence or such words as "datas".

"First, we propose an extension algorithm named Simon k-means, which is a simple k-means based."  K-means based what?

"Our evaluations on several classification show that our design can achieve 5.27X compression and 74.3% multiply-accumulate (MAC) reduction". It is not clear, what is compressed.

"Inspired by this, this paper proposes a model compression scheme based on Simon k-means, which can significantly reduce the number of multiply-accumulate operations and minimize the scale of the model with a slight loss of accuracy". The authors just mention the Simon k-means without any explanation what it is, whose algorithm it is, etc.

Section 2 contains no bibliography references. This is incorrect since this section contains no new results.

Description of the equation (1) is incorrect: centroids are denoted as c_j but described as k_j. Equation (1) is a statement of the optimization problem, however, it is not clear which values must be found (in the k-means problems, centroid coordinates must be found).

" For the convolutional layers and the fully connected layers, the trained weights are clustered through Simon k-means for convolutional layers and Simon k-means for fully connected layers respectively". From this sentence, it is not clear what the difference is. Both convolutional and fully fully connected layers are clustered by the Simon k-means?

"The k-means, which uses randomly selected initial centroids, obviously cannot satisfy our requirements." There are various ways to initialize the k-means algorithm, not necessarily random.

From Section 4.1, it is not clear whether Algorithm 1 is new or not.

In Step 1 of Algorithm 1, weight variable is initialized, however, in step 2, it is assigned sort(x). What the initialization is for? What sort(x) means if x is a matrix? How can we sort a matrix?

From step 5, it is clear that weitght is a vector. How the result of sorting a matrix can be a vector?

THe description of Algorithm 1 is unclear. The same to Algorithm 2. The authors must carefully check all the variable descriptions etc.

After (3) and (4), points of commas are omitted.

From Section 5.2.1, it is not clear what is the accuracy (which measure was used)?

The discussion is too short, and there is no conclusion.

The buibliography is too short.

Reviewer 2 Report

This manuscript proposes a model compression algorithm based on Simon k-means, specifically designed to support a hardware acceleration scheme. Adequate revisions to the following points should be undertaken to justify the recommendation for publication.

  • This paper has more than spelling and grammatical errors. Please fix all of them. And some of the sentences are not understood.
  • The abstract section is very long. Please rewrite an abstract section, justify an obtained result and contribution, improve a proposed method, etc.
  • The authors should clearly state the limitations of the proposed method in other applications.
  • Figures have low quality; please improve them.
  • Please merge the Result and Discussion section and improve this section by comparing new methods
  • Please rewrite the background section
  • Please add the title of the end section (Conclusion and Future Works) and write this section

Good luck

Round 2

Reviewer 1 Report

The authors significantly improved the paper. It can be accepted. English still needs to be slightly improved.

Conclusion section is still to short.

Reviewer 2 Report

In the previous review, another reviewer and I commented on improving the manuscript to become publishable in this journal. Unfortunately, the authors did not pay attention, and they ridiculed the comments. So, in my opinion, it needs to revise. In addition to reviewing and modifying previous comments, consider these as well:

Ø Figure 1 needs to improve and please re-drave it.

Ø Please change the title of the end section (Conclusion) to (Conclusion and Future Works) and write some future work on your proposed method.

Ø Please rewrite an abstract section, justify an obtained result and contribution and how to improve a proposed method, etc.

Good Luck
